# Deep unsupervised learning through spatial contrasting

**Elad Hoffer**
Technion - Israel Institute of Technology
Haifa, Israel
ehoffer@tx.technion.ac.il

**Itay Hubara**
Technion - Israel Institute of Technology
Haifa, Israel
itayh@tx.technion.ac.il

**Nir Ailon**
Technion - Israel Institute of Technology
Haifa, Israel
nailon@cs.technion.ac.il

## Abstract

Convolutional networks have marked their place over the last few years as the best performing model for various visual tasks. They are, however, most suited for supervised learning from large amounts of labeled data. Previous attempts have been made to use unlabeled data to improve model performance by applying unsupervised techniques. These attempts require different architectures and training methods. In this work we present a novel approach for unsupervised training of Convolutional networks that is based on contrasting between spatial regions within images. This criterion can be employed within conventional neural networks and optimized using standard techniques such as SGD and back-propagation, thus complementing supervised methods.

## 1  Introduction

For the past few years convolutional networks (ConvNets, CNNs) LeCun et al. (1998) have proven themselves as a successful model for vision related tasks Krizhevsky et al. (2012) Mnih et al. (2015) Pinheiro et al. (2015) Razavian et al. (2014). A convolutional network is composed of multiple convolutional and pooling layers, followed by a fully-connected affine transformations. As with other neural network models, each layer is typically followed by a non-linearity transformation such as a rectified-linear unit (ReLU).

A convolutional layer is applied by cross correlating an image with a trainable weight filter. This stems from the assumption of stationarity in natural images, which means that parameters learned for one local region in an image can be shared for other regions and images.

Deep learning models, including convolutional networks, are usually trained in a supervised manner, requiring large amounts of labeled data (ranging between thousands to millions of examples per-class for classification tasks) in almost all modern applications. These models are optimized using a variant of stochastic-gradient-descent (SGD) over batches of images sampled from the whole training dataset and their ground truth-labels. Gradient estimation for each one of the optimized parameters is done by back propagating the objective error from the final layer towards the input. This is commonly known as "backpropagation" Rumelhart et al..

In early works, unsupervised training was used as a part of pre-training procedure to obtain an effective initial state of the model. The network was later fine-tuned in a supervised manner as displayed by Hinton (2007). Such unsupervised pre-training procedures were later abandoned, since they provided no apparent benefit over other initialization heuristics in more careful fully supervised training regimes. This led to the de-facto almost exclusive usage of neural networks in supervised environments.

In this work we will present a novel unsupervised learning criterion for convolutional network based on comparison of features extracted from regions within images. Our experiments indicate that by

using this criterion to pre-train networks we can improve their performance and achieve state-of-the-art results.

## 2 PREVIOUS WORKS

Using unsupervised methods to improve performance have been the holy grail of deep learning for the last couple of years and vast research efforts have been focused on that. We hereby give a short overview of the most popular and recent methods that tried to tackle this problem.

**AutoEncoders and reconstruction loss**   These are probably the most popular models for unsupervised learning using neural networks, and ConvNets in particular. Autoencoders are NNs which aim to transform inputs into outputs with the least possible amount of distortion. An Autoencoder is constructed using an encoder $G(x; w_1)$ that maps an input to a hidden compressed representation, followed by a decoder $F(y; w_2)$, that maps the representation back into the input space. Mathematically, this can be written in the following general form:

$$\hat{x} = F(G(x; w_1); w_2)$$

The underlying encoder and decoder contain a set of trainable parameters that can be tied together and optimized for a predefined criterion. The encoder and decoder can have different architectures, including fully-connected neural networks, ConvNets and others. The criterion used for training is the reconstruction loss, usually the mean squared error (MSE) between the original input and its reconstruction Zeiler et al. (2010)

$$min\|x - \hat{x}\|^2$$

This allows an efficient training procedure using the aforementioned backpropagation and SGD techniques. Over the years autoencoders gained fundamental role in unsupervised learning and many modification to the classic architecture were made. Ng (2011) regularized the latent representation to be sparse, Vincent et al. (2008) substituted the input with a noisy version thereof, requiring the model to denoise while reconstructing. Kingma et al. (2014) obtained very promising results with variational autoencoders (VAE). A variational autoencoder model inherits typical autoencoder architecture, but makes strong assumptions concerning the distribution of latent variables. They use variational approach for latent representation learning, which results in an additional loss component which required a new training algorithm called Stochastic Gradient Variational Bayes (SGVB). VAE assumes that the data is generated by a directed graphical model $p(x|z)$ and require the encoder to learn an approximation $q_{w_1}(z|x)$ to the posterior distribution $p_{w_2}(z|x)$ where $w_1$ and $w_2$ denote the parameters of the encoder and decoder. The objective of the variational autoencoder in that case has the following form:

$$\mathcal{L}(w_1, w_2, x) = -D_{KL}(q_{w_1}(z|x)||p_{w_2}(z)) + \mathbb{E}_{q_{w_1}(z|x)}\big(\log p_{w_2}(x|z)\big)$$

Recently, a stacked set of denoising autoencoders architectures showed promising results in both semi-supervised and unsupervised tasks. A stacked what-where autoencoder by Zhao et al. (2015) computes a set of complementary variables that enable reconstruction whenever a layer implements a many-to-one mapping. Ladder networks by Rasmus et al. (2015) - use lateral connections and layer-wise cost functions to allow the higher levels of an autoencoder to focus on invariant abstract features.

**Exemplar Networks:**   The unsupervised method introduced byDosovitskiy et al. (2014) takes a different approach to this task and trains the network to discriminate between a set of pseudo-classes. Each pseudo-class is formed by applying multiple transformations to a randomly sampled image patch. The number of pseudo-classes can be as big as the size of the input samples. This criterion ensures that different input samples would be distinguished while providing robustness to the applied transformations. In this work we will explore an alternative method with a similar motivation.

**Context prediction** Another method for unsupervised learning by context was introduced by Doersch et al. (2015). This method uses an auxiliary criterion of predicting the location of an image patch given another from the same image. This is done by classification to 1 of 9 possible locations. Although the work of Doersch et al. (2015) and ours both use patches from an image to perform unsupervised learning, the methods are quite different. Whereas the former used a classification criterion over the spatial location of each patch within a single image, our work is concerned with comparing patches from several images to each other. We claim that this encourages discriminability between images (which we feel to be important aspect of feature learning), and was not an explicit goal in previous work.

**Adversarial Generative Models:** This a recently introduced model that can be used in an unsupervised fashion Goodfellow et al. (2014). Adversarial Generative Models uses a set of networks, one trained to discriminate between data sampled from the true underlying distribution (e.g., a set of images), and a separate generative network trained to be an adversary trying to confuse the first network. By propagating the gradient through the paired networks, the model learns to generate samples that are distributed similarly to the source data. As shown by Radford et al. (2015),this model can create useful latent representations for subsequent classification tasks.

**Sampling Methods:** Methods for training models to discriminate between a very large number of classes often use a *noise contrasting criterion*. In these methods, roughly speaking, the posterior probability $P(t|y_t)$ of the ground-truth target $t$ given the model output on an input sampled from the true distribution $y_t = F(x)$ is maximized, while the probability $P(t|y_n)$ given a noise measurement $y = F(n)$ is minimized. This was successfully used in a language domain to learn unsupervised representation of words. The most noteworthy case is the word2vec model introduced by Mikolov et al. (2013). When using this setting in language applications, a natural contrasting noise is a smooth approximation of the Unigram distribution. A suitable contrasting distribution is less obvious when data points are sampled from a high dimensional continuous space, such as the case of image patches.

## 2.1 PROBLEMS WITH CURRENT APPROACHES

Only recently the potential of ConvNets in an unsupervised environment began to bear fruit, still we believe it is not fully uncovered.

The majority of unsupervised optimization criteria currently used are based on variations of reconstruction losses. One limitation of this fact is that a pixel level reconstruction is non-compliant with the idea of a discriminative objective, which is expected to be agnostic to low level information in the input. In addition, it is evident that MSE is not best suited as a measurement to compare images, for example, viewing the possibly large square-error between an image and a single pixel shifted copy of it. Another problem with recent approaches such as Rasmus et al. (2015); Zeiler et al. (2010) is their need to extensively modify the original convolutional network model. This leads to a gap between unsupervised method and the state-of-the-art, supervised, models for classification - which can hurt future attempt to reconcile them in a unified framework, as well as efficiently leverage unlabeled data with otherwise supervised regimes.

## 3 LEARNING BY COMPARISONS

The most common way to train NN is by defining a loss function between the target values and the network output. Learning by comparison approaches the supervised task from a different angle. The main idea is to use distance comparisons between samples to learn useful representations. For example, we consider relative and qualitative examples of the form $X_1$ is closer to $X_2$ than $X_1$ is to $X_3$. Using a comparative measure with neural network to learn embedding space was introduced in the "Siamese network" framework by Bromley et al. (1993) and later used in the works of Chopra et al. (2005). One use for this methods is when the number of classes is too large or expected to vary over time, as in the case of face verification, where a face contained in an image has to compared against another image of a face. This problem was recently tackled by Schroff et al. (2015) for training a convolutional network model on triplets of examples. There, one image served as an *anchor* $x$, and an additional pair of images served as a positive example $x_+$ (containing an instance

of the face of the same person) together with a negative example $x_-$, containing a face of a different person. The training objective was on the embedded distance of the input faces, where the distance between the anchor and positive example is adjusted to be smaller by at least some constant $\alpha$ from the negative distance. More precisely, the loss function used in this case was defined as

$$L(x, x_+, x_-) = \max\left\{\|F(x) - F(x_+)\|_2 - \|F(x) - F(x_-)\|_2 + \alpha, 0\right\} \tag{1}$$

where $F(x)$ is the embedding (the output of a convolutional neural network), and $\alpha$ is a predefined margin constant. Another similar model used by Hoffer & Ailon (2015) with triplets comparisons for classification, where examples from the same class were trained to have a lower embedded distance than that of two images from distinct classes. This work introduced a concept of a distance ratio loss, where the defined measure amounted to:

$$L(x, x_+, x_-) = \frac{e^{-\|F(x) - F(x_+)\|_2}}{e^{-\|F(x) - F(x_+)\|_2} + e^{-\|F(x) - F(x_-)\|_2}} \tag{2}$$

This loss has a flavor of a probability of a biased coin flip. By 'pushing' this probability to zero, we express the objective that pairs of samples coming from distinct classes should be less similar to each other, compared to pairs of samples coming from the same class. It was shown empirical by Balntas et al. (2016) to provide better feature embeddings than the margin based distance loss 1

## 4    OUR CONTRIBUTION: SPATIAL CONTRASTING

One implicit assumption in convolutional networks, is that features are gradually learned hierarchically, each level in the hierarchy corresponding to a layer in the network. Each spatial location within a layer corresponds to a region in the original image. It is empirically observed that deeper layers tend to contain more 'abstract' information from the image. Intuitively, features describing different regions within the same image are likely to be semantically similar (e.g. different parts of an animal), and indeed the corresponding deep representations tend to be similar. Conversely, regions from two probably unrelated images (say, two images chosen at random) tend to be far from each other in the deep representation. This logic is commonly used in modern deep networks such as Szegedy et al. (2015) Lin et al. (2013) He et al. (2015), where a global average pooling is used to aggregate spatial features in the final layer used for classification.

Our suggestion is that this property, often observed as a side effect of supervised applications, can be used as a desired objective when learning deep representations in an unsupervised task. Later, the resulting representation can be used, as typically done, as a starting point or a supervised learning task. We call this idea which we formalize below *Spatial contrasting*. The spatial contrasting criterion is similar to noise contrasting estimation Gutmann & Hyvärinen (2010) Mnih & Kavukcuoglu (2013), in trying to train a model by maximizing the expected probability on desired inputs, while minimizing it on contrasting sampled measurements.

### 4.1    FORMULATION

We will concern ourselves with samples of images patches $\tilde{x}^{(m)}$ taken from an image $x$. Our convolutional network model, denoted by $F(x)$, extracts spatial features $f$ so that $f^{(m)} = F(\tilde{x}^{(m)})$ for an image patch $\tilde{x}^{(m)}$. We will also define $P(f_1|f_2)$ as the probability for two features $f_1, f_2$ to occur together in the same image.
We wish to optimize our model such that for two features representing patches taken from the same image $\tilde{x}_i^{(1)}, \tilde{x}_i^{(2)} \in x_i$ for which $f_i^{(1)} = F(\tilde{x}_i^{(1)})$ and $f_i^{(2)} = F(\tilde{x}_i^{(2)})$, $P(f_i^{(1)}|f_i^{(2)})$ will be maximized.
This means that features from a patch taken from a specific image can effectively predict, under our model, features extracted from other patches in the same image. Conversely, we want our model to minimize $P(f_i|f_j)$ for $i, j$ being two patches taken from distinct images. Following the logic presented before, we will need to sample *contrasting patch* $\tilde{x}_j^{(1)}$ from a different image $x_j$ such that $P(f_i^{(1)}|f_i^{(2)}) > P(f_j^{(1)}|f_i^{(2)})$, where $f_j^{(1)} = F(\tilde{x}_j^{(1)})$. In order to obtain contrasting samples, we use regions from two random images in the training set. We will use a distance ratio, described earlier

in Eq. (2) for the supervised case, to represent the probability two feature vectors were taken from the same image. The resulting training loss for a pair of images will be defined as

$$L_{SC}(x_1, x_2) = -\log \frac{e^{-\|f_1^{(1)} - f_1^{(2)}\|_2}}{e^{-\|f_1^{(1)} - f_1^{(2)}\|_2} + e^{-\|f_1^{(1)} - f_2^{(1)}\|_2}} \tag{3}$$

Effectively minimizing a log-probability under the SoftMax measure. This formulation is portrayed in figure 4.1. Since we sample our contrasting sample from the same underlying distribution, we can evaluate this loss considering the image patch as both patch compared (anchor) and contrast symmetrically. The final loss will be the average between these estimations:

$$\widehat{L}_{SC}(x_1, x_2) = \frac{1}{2}\left[L_{SC}(x_1, x_2) + L_{SC}(x_2, x_1)\right]$$

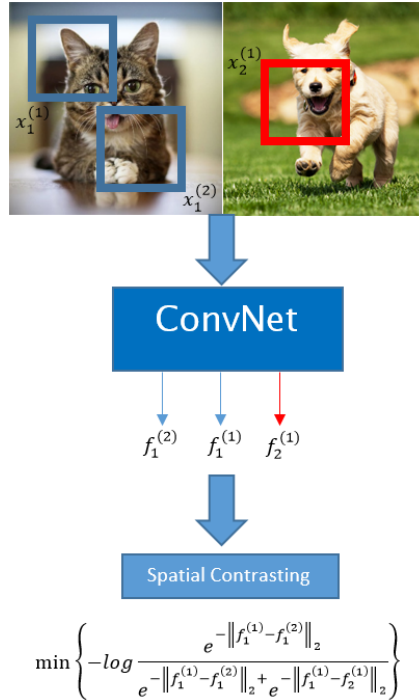

Figure 1: Spatial contrasting depiction.

## 4.2 METHOD

Convolutional network are usually trained using SGD over mini-batch of samples, therefore we can extract patches and *contrasting patches* without changing the network architecture. Each image serves as both anchor and positive patches, for which the corresponding features should be closer, as well as contrasting samples for other images in that batch. For a batch of $N$ images, two samples from each image are taken, and $N^2$ different distance comparisons are made. The final loss is defined as the average distance ratio for all images in the batch:

$$\overline{L}_{SC}(\{x\}_{i=1}^N) = \frac{1}{N}\sum_{i=1}^N L_{SC}(x_i, \{x\}_{j\neq i}) = -\frac{1}{N}\sum_{i=1}^N \log \frac{e^{-\|f_i^{(1)} - f_i^{(2)}\|_2}}{\sum_{j=1}^N e^{-\|f_i^{(1)} - f_j^{(2)}\|_2}} \tag{4}$$

Since the criterion is differentiable with respect to its inputs, it is fully compliant with standard methods for training convolutional network and specifically using backpropagation and gradient descent. Furthermore, SC can be applied to any layer in the network hierarchy. In fact, SC can be used at multiple layers within the same convolutional network. The spatial properties of the

features means that we can sample directly from feature space $\tilde{f}^{(m)} \in f$ instead of from the original image. Therefore SC has a simple implementation which doesn't require substation amount of computation. The complete algorithm for batch training is described in Algorithm (1). Similar to the batch normalization (BN) layer Ioffe & Szegedy (2015), a recent usage for batch statistics in neural networks, SC also uses the batch statistics. While BN normalize the input based on the batch statistics, SC sample from it. This can be viewed as a simple sampling from the space of possible features describing a patch of image.

---

**Algorithm 1** Calculation the spatial contrasting loss

---

**Require:** $X = \{x\}_{i=1}^{N}$ # Training on batches of images

 # Get the spatial features for the whole batch of images
 # Size: $N \times W_f \times H_f \times C$
 $\{f\}_{i=1}^{N} \leftarrow \texttt{ConvNet}(X)$

 # Sample spatial features and calculate embedded distance between all pairs of images
 **for** i = 1 **to** N **do**
 $\tilde{f}_i^{(1)} \leftarrow \texttt{sample}(f_i)$
 **for** j = 1 **to** N **do**
 $\tilde{f}_j^{(2)} \leftarrow \texttt{sample}(f_j)$
 $Dist(i,j) \leftarrow \|\tilde{f}_i^{(1)} - \tilde{f}_j^{(2)}\|_2$
 **end for**
 **end for**

 # Calculate log SoftMax normalized distances
 $d_i \leftarrow -\log \frac{e^{-Dist(i,i)}}{\sum_{k=1}^{N} e^{-Dist(i,k)}}$

 # Spatial contrasting loss is the mean of distance ratios
 **return** $\frac{1}{N} \sum_{i=1}^{N} d_i$

---

## 5 EXPERIMENTS

In this section we report empirical results showing that using SC loss as an unsupervised pretraining procedure can improve state-of-the-art performance on subsequent classification. We experimented with MNIST, CIFAR-10 and STL10 datasets. We used modified versions of well studied networks such as those of Lin et al. (2013) and Rasmus et al. (2015). A detailed description of our architecture can be found in 4.

In each one of the experiments, we used the spatial contrasting criterion to train the network on the unlabeled images. In each usage of SC criterion, patch features were sampled from the preceding layer in uniform. We note that spatial size of sampled patches ranged between datasets, where on STL10 and Cifar10 it covered about $30\%$ of the image, MNIST required the use of larger patches covering almost the entire image.Training was done by using SGD with an initial learning rate of $0.1$ that was decreased by a factor of $10$ whenever the measured loss stopped decreasing. After convergence, we used the trained model as an initialization for a supervised training on the complete labeled dataset. The supervised training was done following the same regime, only starting with a lower initial learning rate of $0.01$. We used mild data augmentations, such as small translations and horizontal mirroring.
The datasets we used are:

- **STL10** (Coates et al. (2011)). This dataset consists of $100,000$ $96 \times 96$ colored, unlabeled images, together with another set of $5,000$ labeled training images and $8,000$ test images . The label space consists of 10 object classes.
- **Cifar10** (Krizhevsky & Hinton (2009)). The well known CIFAR-10 is an image classification benchmark dataset containing $50,000$ training images and $10,000$ test images. The

Table 1: State of the art results on STL-10 dataset

| Model | STL-10 test accuracy |
|---|---|
| Zero-bias Convnets - Paine et al. (2014) | 70.2% |
| Triplet network - Hoffer & Ailon (2015) | 70.7% |
| Exemplar Convnets - Dosovitskiy et al. (2014) | 72.8% |
| Target Coding - Yang et al. (2015) | 73.15% |
| Stacked what-where AE - Zhao et al. (2015) | 74.33% |
| Spatial contrasting initialization (this work) | $81.34\% \pm 0.1$ |
| The same model without initialization | $72.6\% \pm 0.1$ |

image sizes $32 \times 32$ pixels, with color. The classes are airplanes, automobiles, birds, cats, deer, dogs, frogs, horses, ships and trucks.

- **MNIST** (LeCun et al. (1998)). The MNIST database of handwritten digits is one of the most studied dataset benchmark for image classification. The dataset contains 60,000 examples of handwritten digits from 0 to 9 for training and 10,000 additional examples for testing. Each sample is a 28 x 28 pixel gray level image.

All experiments were conducted using the Torch7 framework by Collobert et al. (2011). Code reproducing these results will by available at `https://github.com/eladhoffer/SpatialContrasting`.

## 5.1 RESULTS ON STL10

Since STL10 dataset is comprised of mostly unlabeled data, it is most suitable to highlight the benefits of the spatial contrasting criterion. The initial training was unsupervised, as described earlier, using the entire set of $105,000$ samples (union of the original unlabeled set and labeled training set). The representation outputted by the training, was used to initialize supervised training on the $5,000$ labeled images. Evaluation was done on a separate test set of $8,000$ samples. Comparing with state of the art results, we see an improvement of 7% in test accuracy over the best model by Zhao et al. (2015), setting the SC as best model at $81.3\%$ test classification accuracy (see Table (1)). We note that the results of Dosovitskiy et al. (2014) are achieved with no fine-tuning over labeled examples, which may be unfair to this work. We also compare with the same network, but without SC initialization, which achieves a lower classification of $72.6\%$. This is an indication that indeed SC managed to leverage unlabeled examples to provide a better initialization point for the supervised model.

## 5.2 RESULTS ON CIFAR10

For Cifar10 dataset, we use the same setting as Coates & Ng (2012) and Hui (2013) to test a model's ability to learn from unlabeled images. Here, only $4,000$ samples out of $50,000$ are used with their label annotation, and the rest of the samples can be used only in an unsupervised manner. The final test accuracy is measured on the entire $10,000$ test set.
In our experiments, we trained our model using SC criterion on the entire dataset, and then used only $400$ labeled samples per class (for a total of $4000$) in a supervised regime over the initialized network. The results are compared with previous efforts in Table (2). Using the SC criterion allowed an improvement of 6.8% over a non-initialized model, and achieved a final test accuracy of 79.2%. This is a competitive result with current state-of-the-art models.

## 5.3 RESULTS ON MNIST

The MNIST dataset is very different in nature from the Cifar10 and STL10 datasets, we experimented earlier. The biggest difference, relevant to this work, is that spatial regions sampled from MNIST images usually provide very little, or no information. Thus, SC is much less suited for MNIST dataset, and was conjured to have little benefit. We still, however, experimented with initializing a model with SC criterion and continuing with a fully-supervised regime over all labeled

Table 2: State of the art results on Cifar10 dataset with only 4000 labeled samples

| Model | Cifar10 (400 per class) test accuracy |
|---|---|
| Convolutional K-means Network - Coates & Ng (2012) | 70.7% |
| View-Invariant K-means - Hui (2013) | 72.6% |
| DCGAN - Radford et al. (2015) | 73.8% |
| Exemplar Convnets - Dosovitskiy et al. (2014) | 76.6% |
| Ladder networks - Rasmus et al. (2015) | 79.6% |
| Conv-CatGan Springenberg (2016) | 80.42% ($\pm$ 0.58) |
| ImprovedGan Salimans et al. (2016) | **81.37% ($\pm$ 2.32)** |
| Spatial contrasting initialization (this work) | 79.2%($\pm$0.3) |
| The same model without initialization | 72.4%($\pm$0.1) |

Table 3: results on MNIST dataset

| Model | MNIST test error |
|---|---|
| Stacked what-where AE - Zhao et al. (2015) | 0.71% |
| Triplet network - Hoffer & Ailon (2015) | 0.56% |
| Jarrett et al. (2009) | 0.53% |
| Ladder networks - Rasmus et al. (2015) | 0.36% |
| DropConnect - Wan et al. (2013) | 0.21% |
| Spatial contrasting initialization (this work) | 0.34% $\pm$ 0.02 |
| The same model without initialization | 0.63% $\pm$ 0.02 |

examples. We found again that this provided benefit over training the same network without pre-initialization, improving results from $0.63\%$ to $0.34\%$ error on test set. As mentioned previously, the effective compared patches of MNIST covered almost the entire image area. This can be attributed to the fact that MNIST requires global features to differentiate between digits. The results, compared with previous attempts are included in Table (3).

## 6 CONCLUSIONS AND FUTURE WORK

In this work we presented spatial contrasting - a novel unsupervised criterion for training convolutional networks on unlabeled data. Its is based on comparison between spatial features sampled from a number of images. We've shown empirically that using spatial contrasting as a pretraining technique to initialize a ConvNet, can improve its performance on a subsequent supervised training. In cases where a lot of unlabeled data is available, such as the STL10 dataset, this translates to state-of-the-art classification accuracy in the final model.

Since the spatial contrasting loss is a differentiable estimation that can be computed within a network parallel to supervised losses, in future work we plan to embed it as a semi-supervised model. This usage will allow to create models that can leverage both labeled an unlabeled data, and can be compared to similar semi-supervised models such as the ladder network Rasmus et al. (2015). It is is also apparent that contrasting can occur in dimensions other than the spatial, the most straightforward is the temporal dimension. This suggests that similar training procedure can be applied on segments of sequences to learn useful representation without explicit supervision.

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

# 7 APPENDIX

Table 4: Convolutional models used, based on Lin et al. (2013), Rasmus et al. (2015)

| Model | | |
|---|---|---|
| STL10 | CIFAR-10 | MNIST |
| Input: $96 \times 96$ RGB | Input: $32 \times 32$ RGB | Input: $28 \times 28$ monochrome |
| $5 \times 5$ conv. 64 BN ReLU | $3 \times 3$ conv. 96 BN LeakyReLU | $5 \times 5$ conv. 32 ReLU |
| $1 \times 1$ conv. 160 BN ReLU | $3 \times 3$ conv. 96 BN LeakyReLU | |
| $1 \times 1$ conv. 96 BN ReLU | $3 \times 3$ conv. 96 BN LeakyReLU | |
| $3 \times 3$ max-pooling, stride 2 | $2 \times 2$ max-pooling, stride 2 BN | $2 \times 2$ max-pooling, stride 2 BN |
| $5 \times 5$ conv. 192 BN ReLU | $3 \times 3$ conv. 192 BN LeakyReLU | $3 \times 3$ conv. 64 BN ReLU |
| $1 \times 1$ conv. 192 BN ReLU | $3 \times 3$ conv. 192 BN LeakyReLU | $3 \times 3$ conv. 64 BN ReLU |
| $1 \times 1$ conv. 192 BN ReLU | $3 \times 3$ conv. 192 BN LeakyReLU | |
| $3 \times 3$ max-pooling, stride 2 | $2 \times 2$ max-pooling, stride 2 BN | $2 \times 2$ max-pooling, stride 2 BN |
| $3 \times 3$ conv. 192 BN ReLU | | |
| $1 \times 1$ conv. 192 BN ReLU | | |
| $1 \times 1$ conv. 192 BN ReLU | | |
| Spatial contrasting criterion | | |
| $3 \times 3$ conv. 256 ReLU | $3 \times 3$ conv. 192 BN LeakyReLU | $3 \times 3$ conv. 128 BN ReLU |
| $3 \times 3$ max-pooling, stride 2 | $1 \times 1$ conv. 192 BN LeakyReLU | $1 \times 1$ conv. 10 BN ReLU |
| dropout, $p = 0.5$ | $1 \times 1$ conv. 10 BN LeakyReLU | global average pooling |
| $3 \times 3$ conv. 128 ReLU | global average pooling | |
| dropout, $p = 0.5$ | | |
| fully-connected 10 | | |
| 10-way softmax | | |

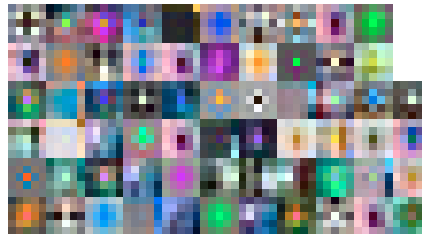

Figure 2: First layer convolutional filters after spatial-contrasting training

