# Peer review of "Deep unsupervised learning through spatial contrasting"

_ICLR 2017 — rejected_

[Official Review · AnonReviewer1 · rating 7 · confidence 4 · 14 Dec 2016]
**No Title**

This paper presents a novel way to do unsupervised pretraining in a deep convolutional network setting (though likely applicable to fully-connected nets as well). The method is that of ‘spatial constrasting’, i.e. of building triplets from patches of input images and learning a presentation that assigns a high score for patches coming from the same image and a low score for patches from diferent images. The method is simple enough that I am surprised that no-one has tried this before (at least according to the previous work in the submission). Here are some comments:


The usage of P(f_i^1 | f_i^2) in Section 4.1 is a bit odd. May be worth defining mathematically what kind of probability the authors are talking about, or just taking that part out (“probability” can be replaced with another word).

I would like to know more about how the method is using the “batch statistics” (end of Section 4.2) by sampling from it, unless the authors simply mean that the just sample from all the possible triples in their batch.

Shouldn’t the number of patches sampled in Algorithm 1 be a hyper-parameter rather than just be 1? Have the authors tried any other value?

I think there are some missing details in the paper, like the patch size or whether the authors have played with it at all (I think this is an important hyper-parameter).

The STL results are quite impressive, but CIFAR-10 maybe not so much. For CIFAR I’d expect that one can try to pre-train on, say, Imagenet + CIFAR to build a better representation. Have the authors considered this?


All in all, this is an interesting piece of work with some obvious applications, and it seems relatively straightforward to implemenent and try. I think I would’ve liked more understanding of what the spatial contrasting actually learns, more empirical studies on the effects of various parameter choices (e.g., patch size) and more attempts at beating the state of the art (e.g. CIFAR).

[Official Review · AnonReviewer3 · rating 6 · confidence 4 · 16 Dec 2016]
**Deep Unsupervised Learning through Spatial Contrasting**

The proposed self supervised loss is formulated using a Siamese architecture and encourages patches from the same image to lie closer in feature space than a contrasting patch taken from a different, random image. The loss is very similar in spirit to that of Doersch et al. ICCV 2015 and Isola et al. ICLR 2016 workshop. It seems that the proposed loss is actually a simplified version of Doersch et al. ICCV 2015 in that it does not make use of the spatial offset, a freely available self supervised signal in natural images. Intuitively, it seems that the self-supervised problem posed by this method is strictly simpler, and therefore less powerful, than that of the aforementioned work. I would like to see more discussion on the comparison of these two approaches. Nevertheless the proposed method seems to be effective in achieving good empirical results using this simple loss. Though more implementation details should be provided, such as the effect of patch size, overlap between sampled patches, and any other important measures taken to avoid trivial solutions.

[Official Review · AnonReviewer2 · rating 5 · confidence 4 · 21 Dec 2016]

This paper proposes an unsupervised training objective based on patch contrasting for visual representation learning using deep neural networks. In particular, the feature representations of the patches from the same image are encouraged to be closer than the those from different images. The distance ratios of positive training pairs are optimized. The proposed method are empirically shown to be effective as an initialization method for supervised training. 

Strengths:

- The training objective is reasonable. In particular, high-level features show translation invariance. 

- The proposed methods are effective for initializing neural networks for supervised training on several datasets. 


Weaknesses:

- The methods are technically similar to the “exemplar network” (Dosovitskiy 2015). Cropping patches from a single image can be taken as a type of data augmentation, which is comparable to the data augmentation of positive sample (the exemplar) in (Dosovitskiy 2015). 

- The paper is experimentally misleading.
The results reported in this paper are based on fine-tuning the whole network with supervision. However, in Table 2, the results of exemplar convnets (Dosovitskiy 2015) is from unsupervised feature learning (the network is not finetuned with labeled samples, and only a classifier is trained upon the features). Therefore, the comparison is not fair. I suspect that exemplar convnets (Dosovitskiy 2015) would achieve similar improvements from fine-tuning; so, without such comparisons (head-to-head comparison with and without fine-tuning based on the same architecture except for the loss), the experimental results are not fully convincing. 

Regarding the comparison to “What-where” autoencoder (Zhao et al, 2015), it will be interesting to compare against it in large-scale settings, as shown by Zhang et al, ICML 2016 (Augmenting Supervised Neural Networks with Unsupervised Objectives for Large-Scale Image Classification). Training an AlexNet is not very time-consuming with latest (e.g., TITAN-X level) GPUs. 

The proposed method seems useful only for natural images where different patches from the same image can be similar to each other.

[Final Decision · Program Chairs · 06 Feb 2017]
**ICLR committee final decision**

The paper proposes a formulation for unsupervised learning of ConvNets based on the distance between patches sampled from the same and different images. The novelty of the method is rather limited as it's similar to [Doersch et al. 2015] and [Dosovitsky et al. 2015]. The evaluation is only performed on the small datasets, which limits the potential impact of the contribution.